# A Novel LSTM Model with Interaction Dual Attention for Radar Echo Extrapolation

**Chuyao Luo** **, Xutao Li \*, Yongliang Wen, Yunming Ye and Xiaofeng Zhang**

The Department of Computer Science, Harbin Institute of Technology, Shenzhen 518055, China; luochuyao.dalian@gmail.com (C.L.); yongliangwen.hit@gmail.com (Y.W.); yeyunming@hit.edu.cn (Y.Y.); zhangxiaofeng@hit.edu.cn (X.Z.)
**\*** Correspondence: lixutao@hit.edu.cn; Tel.: +86-8618-5559

**Abstract:** The task of precipitation nowcasting is significant in the operational weather forecast. The radar echo map extrapolation plays a vital role in this task. Recently, deep learning techniques such as Convolutional Recurrent Neural Network (ConvRNN) models have been designed to solve the task. These models, albeit performing much better than conventional optical flow based approaches, suffer from a common problem of underestimating the high echo value parts. The drawback is fatal to precipitation nowcasting, as the parts often lead to heavy rains that may cause natural disasters. In this paper, we propose a novel interaction dual attention long short-term memory (IDA-LSTM) model to address the drawback. In the method, an interaction framework is developed for the ConvRNN unit to fully exploit the short-term context information by constructing a serial of coupled convolutions on the input and hidden states. Moreover, a dual attention mechanism on channels and positions is developed to recall the forgotten information in the long term. Comprehensive experiments have been conducted on CIKM AnalytiCup 2017 data sets, and the results show the effectiveness of the IDA-LSTM in addressing the underestimation drawback. The extrapolation performance of IDA-LSTM is superior to that of the state-of-the-art methods.

**Keywords:** precipitation nowcasting; radar echo extrapolation; deep learning

## 1. Introduction

Precipitation nowcasting refers to predicting the future rainfall intensity within a relatively short period (e.g., 0∼6 h) based on the observation of radar. The prediction is significant for alerting natural disasters caused by heavy rain and guiding the travel plan of people. The key part of the task is the radar echo map extrapolation, namely predicting the radar echo map sequences in the future based on the historical observations. Once the extrapolation is obtained, precipitation nowcasting can be easily obtained with many methods such as the Z-R relationship [1].

Existing radar echo map extrapolation methods can be mainly classified into two types: (1) optical flow-based models [2,3] and (2) deep learning-based algorithms [4–8]. The former calculates a motion field between the adjacent maps based on the assumption that the brightness of pixels is constant. Then, the extrapolation can be made by applying the motion field iteratively. However, the intensity of the echo map always keeps changing, for example, strengthening or weakening. Therefore, the constant assumption on brightness is unreasonable. Moreover, the generation of movement only uses a few recent radar images, which suggests the type of methods cannot utilize the valuable historical observations. The latter builds a mapping from previous radar observations to future echo maps by constructing a neural network, for example, the convolution neural network (CNN) [9,10], recurrent neural network (RNN) [11], and the Spatial Transformer Networks (STN) [12]. One of the representative approaches is the ConvLSTM . The method combines the Convolution (CNN) and Long Short-Term Memory (LSTM). Here, the LSTM captures the temporal dynamics of the hidden states into the temporal memory. CNN is responsible for extracting

spatial information. The deep learning extrapolation approaches usually perform better than the optical flow methods [13], because they do not have the unreasonable constant assumption and they can effectively leverage the valuable historical observations.

However, there is a fatal problem for almost all deep learning-based methods, namely the high echo value part is often underestimated, as shown in Figure 1. We observe that the high radar echo region of the prediction has the trend of disappearing. This phenomenon is universal for other deep learning-based models, which will lead to a serious negative influence on the prevention of disasters caused by strong rainfall. It can be attributed to the following two primary reasons. First, in each step of operation, the various gates in Convolutional Recurrent Neural Network (ConvRNN) are generated by independent convolutions on input and hidden state and a sum fusion. The independent convolution has a limitation because the input and hidden state do not help each other to identify and preserve important information. Consequently, the ConvRNN may lose short-term dependency information in each step. Second, LSTM has a forgetting mechanism to control whether discarding information from previous temporal memory and this process is irreversible [14]. Therefore, in terms of the long-period, the representation of high echo value cannot be found once it is forgotten by temporal memory. That is, the long-term dependency is not nicely modeled.

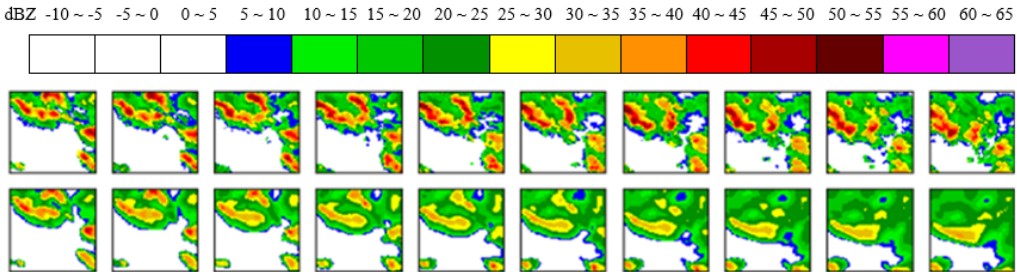

**Figure 1.** An instance of radar echo map prediction. The first line denotes the ground truth and the second line is the prediction by the ConvLSTM model. (Best view in color).

Recently, some methods were proposed to address the issues. First, to preserve the short-term dependency, Wang. et al. proposed PredRNN [15] and PredRNN++ [16] by introducing a spatial memory into the original ConvLSTM. The spatial memory can preserve the spatial information from the bottom to the top layer. Also, Tran., et al. [17] showed that it can be applied in radar echo extrapolation and deliver better performance than ConvLSTM. However, the extra spatial memory does not help the input or the hidden state to select important features because they are convolved independently. Secondly, to preserve the long-term spatiotemporal correlation, Eidetic 3D LSTM(E3D-LSTM) [18] utilizes the self-attention [19,20] module. It can preserve more spatiotemporal representation because the information can be found from historical memories by the attention mechanism. Nevertheless, it merely uses the single spatial attention mechanism to recall the previous temporal memories. The channel correlations are not modeled [21].

To overcome the limitations of existing models, we propose a novel Interactional Dual Attention Long Short-term Memory (IDA-LSTM) by adding (1) an interaction scheme between the hidden state and the input to preserve more import representation, and (2) a dual attention module for both channel and temporal information to obtain better long-term spatiotemporal representation. For the interaction part, we develop a coupled convolution framework for the input and hidden state. In the framework, the input and hidden state are interacted and fused by a serial of coupled convolutions. As a result, a novel input and hidden state are formed, where the important information are selected with the coupled convolutions. For the dual attention part, we combine channel attention and spatial attention module to substitute for the update of the temporal memory at the forget gate. The single spatial attention module used in E3D-LSTM only considers how to selectively

reorganize the feature at each position. However, it ignores the correlation between the different channels. Therefore, channel attention is introduced to further improve the representation in the long term. The experimental results show that the proposed IDA-LSTM is valid to improve the accuracy of precipitation nowcasting, especially in the high echo region. The contribution of our method can be summarized as follows:

1. We first develop the interaction scheme to enhance the short-term dependency modeling ability of ConvRNN approaches. The interaction scheme is a general framework, which can be applied in any ConvRNN model.
2. We introduce the dual attention mechanism to combine the long-term temporal and channel information for the temporal memory cell. The mechanism helps recall the long-term dependency and form better spatiotemporal representation.
3. By applying the interaction scheme and the dual attention mechanism, we propose our IDA-LSTM approach for radar echo map extrapolation. Comprehensive experiments have been conducted. The IDA-LSTM achieves state-of-the-art results, especially in the high radar echo region, on the CIKM AnalytiCup 2017 radar datasets. To reproduce the results, we release the source code at: https://github.com/luochuyao/IDA_LSTM.

## 2. Proposed Method

In this section, we present the proposed IDA-LSTM model. We first introduce the interaction framework which can be applied in any ConvRNN model. Then, we elaborate the dual attention mechanism and describe how to embed the dual attention mechanism and interaction into our model. Finally, we introduce the whole architecture of the proposed IDA-LSTM model.

### 2.1. Interaction Framework

The original ConvRNN models independently apply a convolution into the input and hidden state respectively. This process cannot effectively model the correlations between the input and hidden state. To address it we propose a novel interaction scheme by constructing a serial of coupled convolutions as shown in Figure 2. We assume that the original input and hidden state are $x^0$ and $h^0$ in ConvRNN models. The new input $x^1$ is obtained by the convolutions on $x^0$ and $h^0$ respectively and adding them together. Upon the summation result, a rectified linear threshold unit (ReLU) is appended. Similar to $x^1$, the new hidden state $h^1$ is obtained according to $x^1$ and $h^0$. The process can be repeated for $i$ times to obtain the new hidden state $h^i$ and $x^i$. The new hidden state $h^i$ and $x^i$ will be fed into the ConvRNN unit. Formally, the operation of $h^i$ and $x^i$ in $i$th iterator can be expressed as the following equation:

$$
\begin{aligned}
x^i &= relu(x^{i-1} * W_{xx} + h^{i-1} * W_{hx}) \\
h^i &= relu(x^i * W_{xh} + h^{i-1} * W_{hh})
\end{aligned}
\tag{1}
$$

Here, '$*$' denotes the 2D convolution. From this equation, we can see that each update of the input always uses the information of the hidden state, which means it merges the representation of the hidden state. Besides, the $h^{i-1} * W_{hx}$ also can help the old input $x^{i-1}$ to leverage significant representation by convolution. The same advantage also can be reflected in the hidden state. It aggregates the information from the input and can extract better representation under the guidance of the input. By repeating this process, the formed input and the hidden state can nicely exploit the context information before going into the ConvRNN unit.

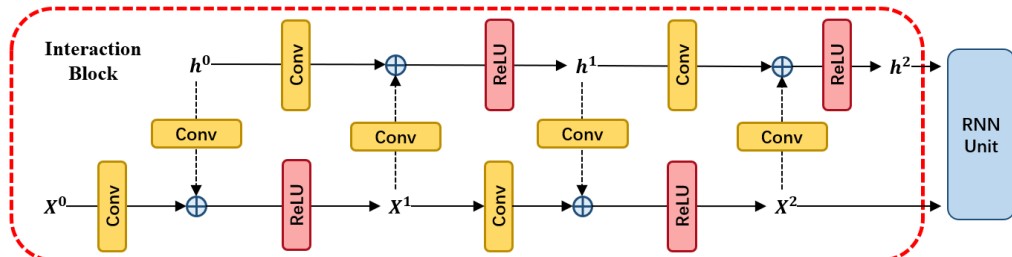

**Figure 2.** The interaction block. It interacts with the representation between the hidden state $h$ and the input $x$.

To examine the performance of the interaction method, we depict Figure 3 which shows the eight one-hour prediction samples $w.$ (with) and $w/o.$ (without) interaction. We utilize the black box to mark the areas with high reflective to emphasize these parts. We can see that the model with the interaction mechanism can effectively improve the nowcasting, especially in the high rainfall regions. For the models without interaction, it even cannot generate the red parts. It implies the interaction scheme is more effective in predicting the high echo value parts. We will further validate its effectiveness in Section 3. We note that this scheme is a general framework which can be applied in any ConvRNN models.

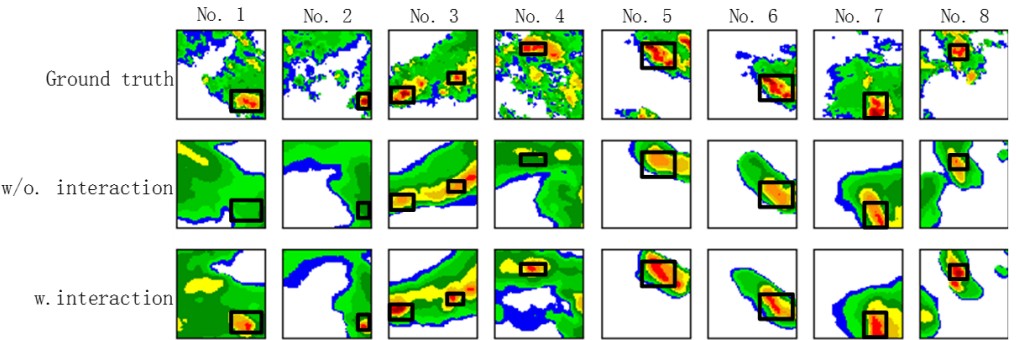

**Figure 3.** The first line is the eight ground truth examples and the rest are predictions after an hour without (second row) interaction and with (the last row). The color scheme is based on the color code in the right part (Best view in color).

*2.2. Dual Attention Mechanism*

To further model the long-range dependency, we develop a dual attention mechanism for both the spatial and channel module, shown as in Figure 4. Next, we detail the dual attention mechanism.

Spatial Attention Module Given the feature map $f_t \in R^{N \times C \times H \times W}$ and it can be regarded as the query $Q_s$, where $N$, $C$, $H$ and $W$ denote the batch size, channel, height and width of the feature map respectively, we directly reshape it to $Q_s \in R^{N \times (H*W) \times C}$. For a series of feature maps $C_{t-\tau:t-1} \in R^{N \times C \times \tau \times H \times W}$, it can be seen as the key $K_s$ and value $V_s$. Here, the $\tau$ is length of the series. Similarly, it can be reshaped into $K_s \in R^{N \times (\tau*H*W) \times C}$ and $V_s \in R^{N \times (\tau*H*W) \times C}$. Next, according to the Equation (2), we can obtain the output of spatial attention:

$$
\begin{aligned}
A_s &= Attn_s(f_t, C_{t-\tau:t-1}) \\
&= norm(C_{t-1} + softmax(Q_s \cdot K_s^T) \cdot V_s) \\
Q_s &= f_t; K_s = V_s = C_{t-\tau:t-1},
\end{aligned}
\tag{2}
$$

which is shown as the green part in Figure 4. Here, the $softmax(Q_s \cdot K_s^T) \in R^{N \times H*W \times \tau*H*W}$ denotes to apply a softmax layer to the matrix product operation of $Q_s$ and $K_s$. It represents position similarity between the $Q_s$ and $K_s$. The value implies the degree of correlation of

given feature maps $f_t$ and the long-term series maps $C_{t-\tau:t-1}$. Then, it can be regarded as the weight to update spatial information by the calculation of matrix product with $V_s$, which selectively integrates the location information from $C_{t-\tau:t-1}$ into the final result $A_s$ in terms of the spatial similarity. Finally, we rebuild the shape of $A_s \in R^{N \times C \times H \times W}$ to let the $f_t$ and $C_{t-1}$ as the output of the module.

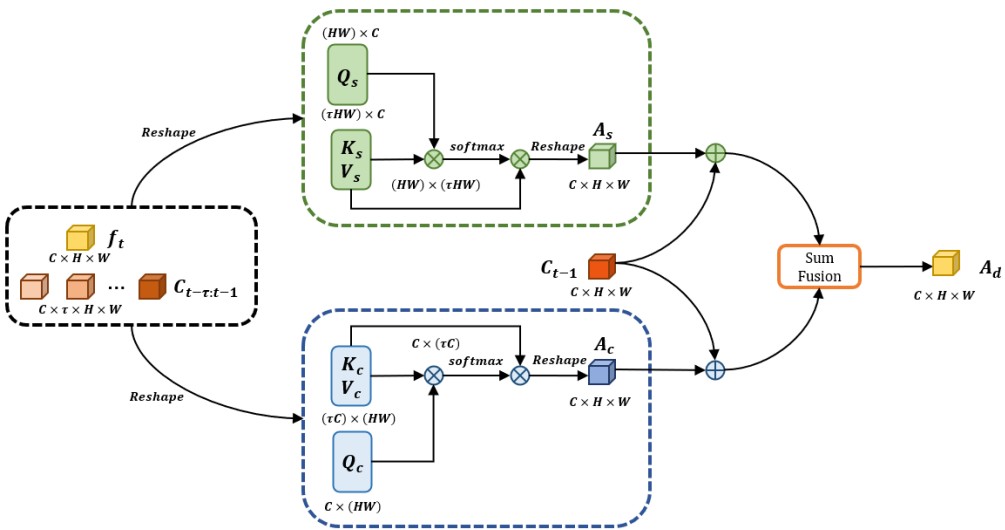

**Figure 4.** The dual attention mechanism embeding in proposed model.

Channel Attention Module aims to model the correlations between the channels. Different from the spatial attention module, the query $f_t \in R^{N \times C \times H \times W}$ are transformed and reshaped to the query $Q_c \in R^{N \times C \times (H*W)}$. Also, the series of feature maps $C_{t-\tau:t-1} \in R^{N \times C \times \tau \times H \times W}$ can be reshaped into key $K_c \in R^{N \times \tau * C \times (H*W)}$ and value $V_c \in R^{N \times (\tau * C) \times (H*W)}$. The channel attention can be expressed as the following equation:

$$
\begin{aligned}
A_c &= Attn_c(f_t, C_{r-\tau:t-1}) \\
&= norm(C_{t-1} + softmax(Q_c \cdot K_c^T) \cdot V_c). \\
Q_c &= f_t; K_c = V_c = C_{t-\tau:t-1}.
\end{aligned}
\tag{3}
$$

Here, the $softmax(Q_c \cdot K_c^T) \in R^{N \times C \times (\tau * C)}$ denotes the query $f_t$'s impact representation on the key $C_{t-\tau:t-1}$ in terms of channels. Then, we perform matrix product between it and the value $V_c$. In the same way, we reshape the result of Equation (3) into appropriate sizes shown as the blue part in Figure 4 and then sum it with the spatial attention result.

Sum Fusion finally integrates the output from the two attention modules. Figure 5 shows the structure of the part in detail. Specifically, it independently applies two convolution layers on $A_s$ and $A_c$ respectively. The first convolution layer involves a convolution with 3 kernel size, a layer normalization and the activate function of ReLU. The second utilizes a convolution operation where kernel size is 1. Then, the element-wise sum is performed on the two results. At last, a convolution layer is applied to generate the final result.

The combination of the results from both parts not only involves the local spatial features but also includes the channel relationship at corresponding position in the long period of $\tau$. Therefore, the output of the dual attention module is more effective than each single module. In summary, the dual attention module can be shown as the following formula:

$$
\begin{aligned}
A_d &= Attn_d(f_t, C_{t-\tau:t-1}) \\
&= SumFusion(Attn_s(f_t, C_{r-\tau:t-1}), Attn_c(f_t, C_{r-\tau:t-1})).
\end{aligned}
\tag{4}
$$

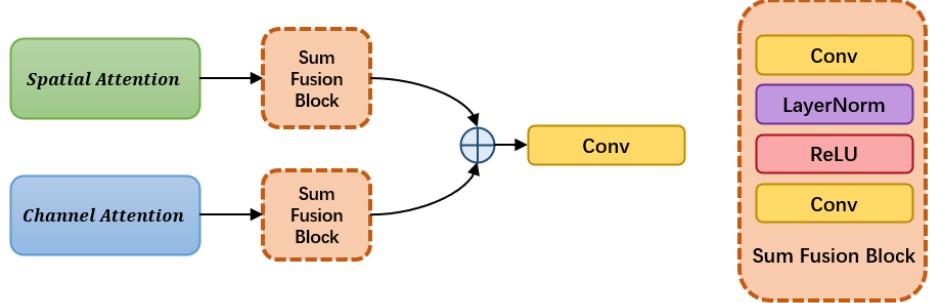

**Figure 5.** The structure of SumFusion modular.

Figure 6 shows the eight one-hour prediction examples with the position attention module, channel attention module and dual attention module, respectively. We can see the channel attention module and spatial attention model cannot accurately predict the high echo value regions. Both of them underestimate the parts. As for the dual attention module, its prediction is more accurate and the high echo value region parts are not underestimated, which implies that the combination of the two attention modules is very effective. We will further demonstrate its advantages in Section 3.

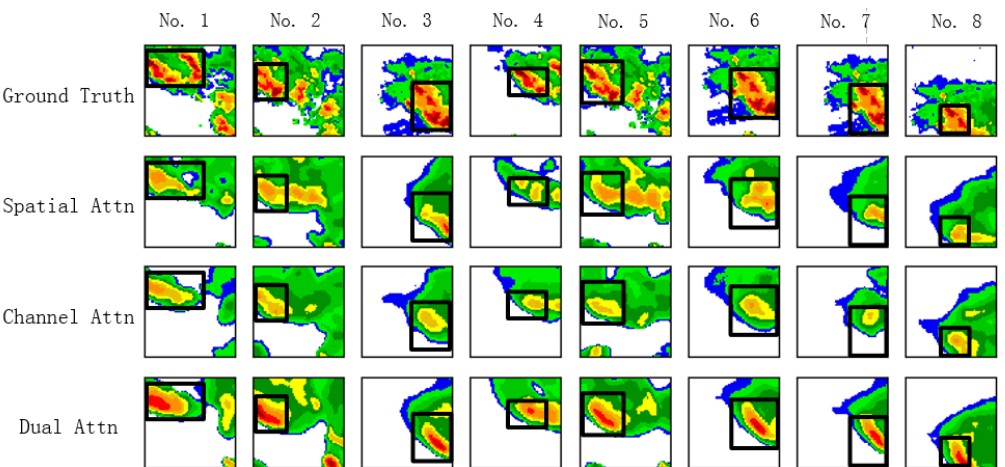

**Figure 6.** The first line is the eight ground truth examples and the rest are predictions after an hour merely with spatial attention (second row), channel attention (third row), and dual attention (the last row). (Best view in color).

### 2.3. The IDA-LSTM Unit

In this subsection, we will introduce how to embed the dual attention mechanism into the LSTM unit to form our IDA-LSTM. Figure 7 illustrates the inside structure of our proposed IDA-LSTM unit. We can see that the input of IDA-LSTM block contains the current input $X_t$, spatial memory $M_t^{l-1}$, historical temporal memories $C_{t-\tau:t-1}^l$ and hidden state $H_{t-1}^l$. The current input $X_t$ and hidden state $H_{t-1}^l$ are first transformed by the developed interaction block. The resulting new input and hidden state are then combined with $C_{t-\tau:t-1}^l$ and as the input of the developed dual attention mechanism to update the temporal memory $C_t^l$. By the dual attention on the multiple previous temporal memories $C_{t-\tau:t-1}^l$, the $C_t^l$ can recover the forgotten information. Finally, the temporal memory $C_t^l$ is delivered to the next time. In addition to temporal memory, in our IDA-LSTM unit, we follow PredRNN and introduce the spatial memory $M_t^l$. Its update scheme is the same as that in the PredRNN. Formally, the calculation of the IDA-LSTM unit is expressed as:

$$i_t = \sigma(W_{xi} * X_t + W_{hi} * H_{t-1}^l + b_i),$$
$$g_t = tanh(W_{xg} * X_t + W_{hg} * H_{t-1}^l + b_g),$$
$$f_t = \sigma(W_{xf} * X_t + W_{hf} * H_{t-1}^l + b_f),$$
$$i_t' = \sigma(W_{xi}' * X_t + W_{mi} * M_t^{l-1} + b_i'),$$
$$g_t' = tanh(W_{xg}' * X_t + W_{mg} * M_t^{l-1} + b_g'),$$
$$f_t' = \sigma(W_{xf}' * X_t + W_{mf} * M_t^{l-1} + b_f'),$$
$$C_t^l = i_t \circ g_t + Attn_d(f_t, C_{t-\tau:t-1}),$$
$$M_t^l = i_t' \circ g_t' + f_t' \circ M_t^{l-1},$$
$$o_t = \sigma(W_{xo} * X_t + W_{ho} * H_{t-1}^l + W_{co} * C_t^l + W_{mo} * M_t^l + b_o),$$
$$H_t^l = o_t \circ tanh(W_{1\times1} * [X_t^l, M_t^l]),$$

$$(5)$$

where $\circ$ is the matrix product and $\tau$ is the number of previous temporal memories. The $*$ denotes the 2D convolution and the *LayerNorm* is layer normalization [22] that is designed to stabilize the training process. $Attn_d$ in Equation (5) denotes the dual attention module in Section 2.2. Here, the forget gate $f_t$ is the query, and the historical memories $C_{t-\tau:t-1}$ denotes the key and value. The function of this mechanism is to control where and what information to emphasize on previous memories so as to produce good predictions. Obviously, the dual attention mechanism can nicely model the long-term dependency.

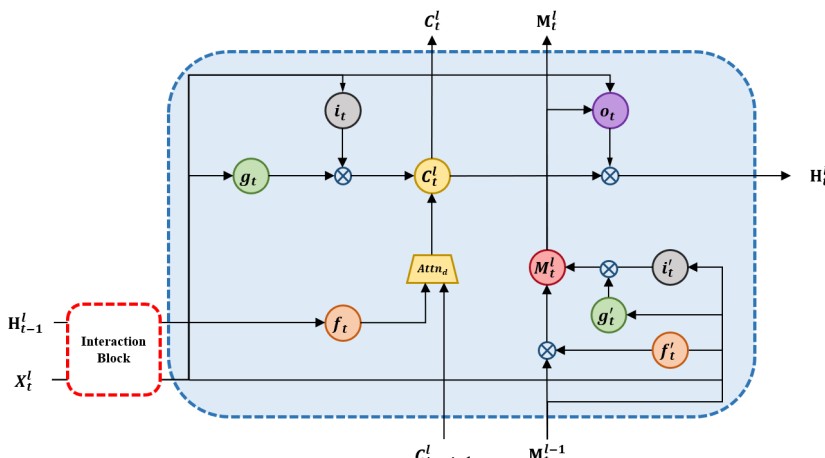

**Figure 7.** The inside structure of Interactional Dual Attention Long Short-term Memory (IDA-LSTM) unit.

### 2.4. The IDA-LSTM Extrapolation Architecture

The architecture of our model is similar to the convolutional recurrent network such as PredRNN [15], PredRNN++ [16] , and Eidetic 3D LSTM [18] model as shown in Figure 8. Our architecture is a four-layer network built upon the IDA-LSTM units. In the architecture, the temporal memory information is delivered along the horizontal direction (shown as black dot lines) and the spatial memory information is transmitted in an zigzag manner (shown as red dot lines). The prediction $\hat{X}_t$ is generated from the output of the top layer after going throw a convolution layer.

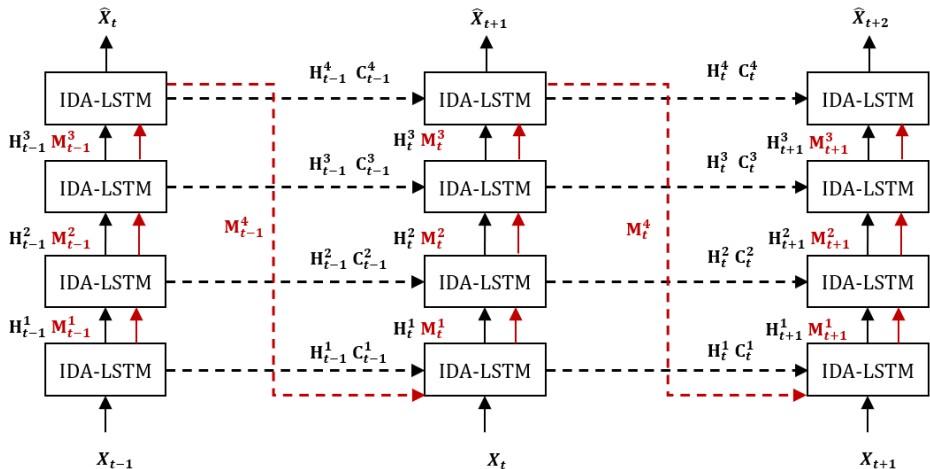

**Figure 8.** The architecture of our model.

## 3. Experiment

### 3.1. Experimental Setup

In this part, we show the introduction of the experiment including the dataset, evaluation metrics, parameter and training setting.

#### 3.1.1. Dataset

We utilize the Conference on Information and Knowledge Management (CIKM) AnalytiCup 2017 challenge dataset to evaluate our model. It contains 10,000 samples as the training set and 4000 samples as the testing set. The sample time of both is different, the data in the training set come from two consecutive years and the test-set is sampled within the next year. We randomly select 2000 samples from the training set as the validation set. Each sample includes 15 CAPPI radar images with an interval of 6 min and elevation-angles (0.5, 1.5, 2.5 and 3.5 km). In this paper, we select the 3.5 km level images to train and test our models. In each sample, the first five echo maps are treated as input and the last ten as the expected output. That is, we aim to predict the one-hour extrapolation based on the half-an-hour observations in the past. Each image covers 101 km × 101 km square with the size of 101 × 101 (pixel). Each pixel represents a resolution of 1 km × 1 km and the original range of it is [0, 255]. We give the Figure 9 to show the distribution of pixels of train-sets, validation-set and test-set. The distribution gap between the training-sets and test implies the challenge of this nowcasting task.

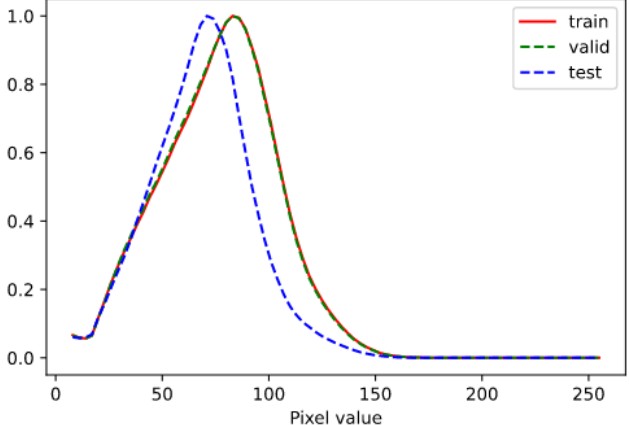

**Figure 9.** The histograms of non-zeros pixel values in the train-sets, validation-set and test-set respectively.

### 3.1.2. Evaluation Metrics

As a preprocessing step, we convert the pixel value of each pixel value as follows:

$$dBZ = pixel\_value \times 95/255 - 10. \tag{6}$$

As for evaluation, we convert the prediction echo map and the ground truth one by thresholding. If the value larger than the given threshold, the corresponding value is set to 1; otherwise it is set to 0. Then we calculate the number of positive predictions TP (prediction = 1, truth = 1), false-positive predictions FP (predictio = 1, truth = 0), true negative predictions TN (prediction = 0, turth = 0) and false-negative predictions FN (predition = 0, truth = 1). Specifically, we use three thresholds namely 5 dBZ, 20 dBZ and 40 dBZ. Finally, we compute the Heidke Skill Score (HSS) [23] and Critical Success Index (CSI) metrics to evaluation the results:

$$HSS = \frac{2(TP \times TN - FN \times FP)}{(TP + FN)(FN + TN) + (TP + FP)(FP + TN)}$$
$$CSI = \frac{TP}{TP + FN + FP}. \tag{7}$$

Moreover, we also apply MAE and SSIM to evaluate our model from a different view.

### 3.1.3. Parameter and Training Setting

The parameter configuration details of IDA-LSTM model are described in Table 1. Here, 'In Kernel', 'In Stride', and 'In Pad' denotes the kernel size, stride, and the padding in the input-to-state 3D convolution respectively. 'State Ker.' and 'Spatial Ker.' denotes the kernel size of the state-to-state and spatial memory 3D convolution respectively. Its stride and padding setting is the same as 'In Stride' and 'In Pad'. For the interaction part as Figure 2 shown, each convolution block applies the convolution kernel with 32 filters, $5 \times 5$ kernel size, 1 stride and the same padding. Besides, each input with shape (101, 101, 1) is padded with zeros into the shape (128, 128, 1) and then is patched to the shape (32, 32, 16). Eventually, at any time step, the output with the shape (32, 32, 16) in the top layer transforms the shape to (128, 128, 1) as the final prediction.

**Table 1.** The details of the IDA-LSTM model . The output of the Layer 5 will be transform the prediction with same shape of input.

| Layer | In Kernel | In Stride | Pad | State Ker. | Spatial Ker. | Ch I/O | In Res | Out Res | Type |
|---|---|---|---|---|---|---|---|---|---|
| Layer 1 | $5 \times 5$ | $1 \times 1$ | $2 \times 2$ | $5 \times 5$ | $5 \times 5$ | 16/32 | $32 \times 32$ | $32 \times 32$ | IDA-LSTM |
| Layer 2 | $5 \times 5$ | $1 \times 1$ | $2 \times 2$ | $5 \times 5$ | $5 \times 5$ | 32/32 | $32 \times 32$ | $32 \times 32$ | IDA-LSTM |
| Layer 3 | $5 \times 5$ | $1 \times 1$ | $2 \times 2$ | $5 \times 5$ | $5 \times 5$ | 32/32 | $32 \times 32$ | $32 \times 32$ | IDA-LSTM |
| Layer 4 | $5 \times 5$ | $1 \times 1$ | $2 \times 2$ | $5 \times 5$ | $5 \times 5$ | 32/32 | $32 \times 32$ | $32 \times 32$ | IDA-LSTM |
| Layer 5 | $1 \times 1$ | $1 \times 1$ | $0 \times 0$ | - | - | 32/16 | $32 \times 32$ | $32 \times 32$ | Conv2D |

Before training, all radar echo maps were normalized to $[-1, 1]$ as the input. our model is optimized with an L1+L2 loss. In the training step, all models are trained by utilizing the ADAM optimizer [24] with the 0.001 learning rate. The batch size of each iteration process and the maximum number of iterations is set to 4 and 80,000 respectively. Besides, the early-stopping strategy was applied. All experiments are implemented in Pytorch and executed on NVIDIA TITAN GPU.

### 3.2. Experimental Results

Table 2 shows the results of all the methods. Here, ConvLSTM, ConvGRU, TrajGRU and PredRNN were previously applied and tested on the data set. From this table, DA-LSTM denotes our model with the dual interaction module but without the interaction part. We can see that the proposed model including DA-LSTM and IDA-LSTM achieves the best performance in terms of the HSS, CSI at all thresholds. Particularly, the evaluated metrics reach 0.2262 and 0.1287 when the threshold is at 40 dBZ, which is 21.74% and 24.47% higher than the second-rank algorithm (MIM model), respectively. It implies that the developed dual attention and interaction modules are helpful for the high rainfall region prediction, which is especially important to alert significant threats on human activity and economy. For the MIM model, it delivers the the second result at two relatively high thresholds (20 dBZ and 40 dBZ). This can be attributed to its memory in memory scheme, which can help to preserve the high echo value regions. Besides, the results of PredRNN, PredRNN++ and TrajGRU are superior to ConvLSTM and ConvGRU. E3DLSTM performs the worst among all the methods.

**Table 2.** Comparison results of ablation study on the CIKM AnalytiCup 2017 competition dataset in terms of Heidke Skill Score (HSS) and Critical Success Index (CSI). Bold denotes the best evaluate index among all models.

| dBZ Threshold | HSS ↑ | | | | CSI ↑ | | | | MAE ↓ | SSIM ↑ |
|---|---|---|---|---|---|---|---|---|---|---|
| | **5** | **20** | **40** | **avg** | **5** | **20** | **40** | **avg** | | |
| ConvLSTM [4] | 0.7035 | 0.4819 | 0.1081 | 0.4312 | 0.7656 | 0.4034 | 0.0578 | 0.4089 | 15.06 | 0.2229 |
| ConvGRU [6] | 0.6776 | 0.4766 | 0.1510 | 0.4351 | 0.7473 | 0.3907 | 0.0823 | 0.4068 | 16.27 | 0.1370 |
| TrajGRU [6] | 0.6828 | 0.4862 | 0.1621 | 0.4437 | 0.7499 | 0.4020 | 0.0888 | 0.4136 | 15.99 | 0.1519 |
| PredRNN [15] | 0.7080 | 0.4911 | 0.1558 | 0.4516 | 0.7691 | 0.4048 | 0.0839 | 0.4198 | 14.54 | 0.3341 |
| PredRNN++ [16] | 0.7075 | 0.4993 | 0.1574 | 0.4548 | 0.7670 | 0.4137 | 0.0862 | 0.4223 | 14.51 | 0.3357 |
| E3D-LSTM [18] | 0.7111 | 0.4810 | 0.1361 | 0.4427 | 0.7720 | 0.4060 | 0.0734 | 0.4171 | 14.78 | 0.3089 |
| MIM [8] | 0.7052 | 0.5166 | 0.1858 | 0.4692 | 0.7628 | 0.4279 | 0.1034 | 0.4313 | 14.69 | 0.2123 |
| DA-LSTM | **0.7184** | 0.5251 | 0.2127 | 0.4854 | **0.7765** | **0.4376** | 0.1202 | 0.4448 | 14.10 | 0.3479 |
| IDA-LSTM | 0.7179 | **0.5264** | **0.2262** | **0.4902** | 0.7752 | 0.4372 | **0.1287** | **0.4470** | **14.09** | **0.3506** |

We draw Figure 10 to describe the HSS and CSI curves of all models at all nowcasting lead time stamps. We can see that our models always keep top positions at all thresholds and any time, which demonstrates the robust superiority of our approach. It is worth pointing out that the gaps between our IDA-LSTM model, and other models are most obvious at the 40 dBZ, which demonstrates our method significantly improves the high rainfall region prediction. Besides, the MIM model delivers the second best performance due to the design of memory in memory. Moreover, the result of PredRNN is always worse than the proposed models, particularly at the high threshold. It implies the effectiveness of the proposed interaction and dual attention mechanisms, because the two parts are the only differences between PredRNN and our IDA-LSTM. As for ConvGRU and ConvLSTM model, they deliver the worst performance.

To better compare and understand the results, we visualize some prediction examples produced by different methods in Figure 11. We can see that only PredRNN, DA-LSTM, and IDA-LSTM can preserve the high echo value regions (red parts) in the 10th prediction image. Among the three methods, IDA-LSTM is the best, followed by DA-LSTM and then PredRNN. The reason is that IDA-LSTM applies both the interaction and dual attention schemes, which better model the short-term and long-term dependency. DA-LSTM applies only the dual attention module. Hence, the performance degenerates a little bit. As for PredRNN, it does not have the interaction and dual attention modules, but only leverages the spatial memory cell. Thus, it can only preserve parts of the high echo value regions. Moreover, we can see from the ground-truth sequence that the high echo value regions increase and the intensity becomes higher as the lead time goes on. The proposed IDA-LSTM can nicely predict the trend. As for other deep learning models, they cannot predict

the high echo value regions and the red parts gradually disappear as the lead time goes on. The fact further confirms our motivation of this paper.

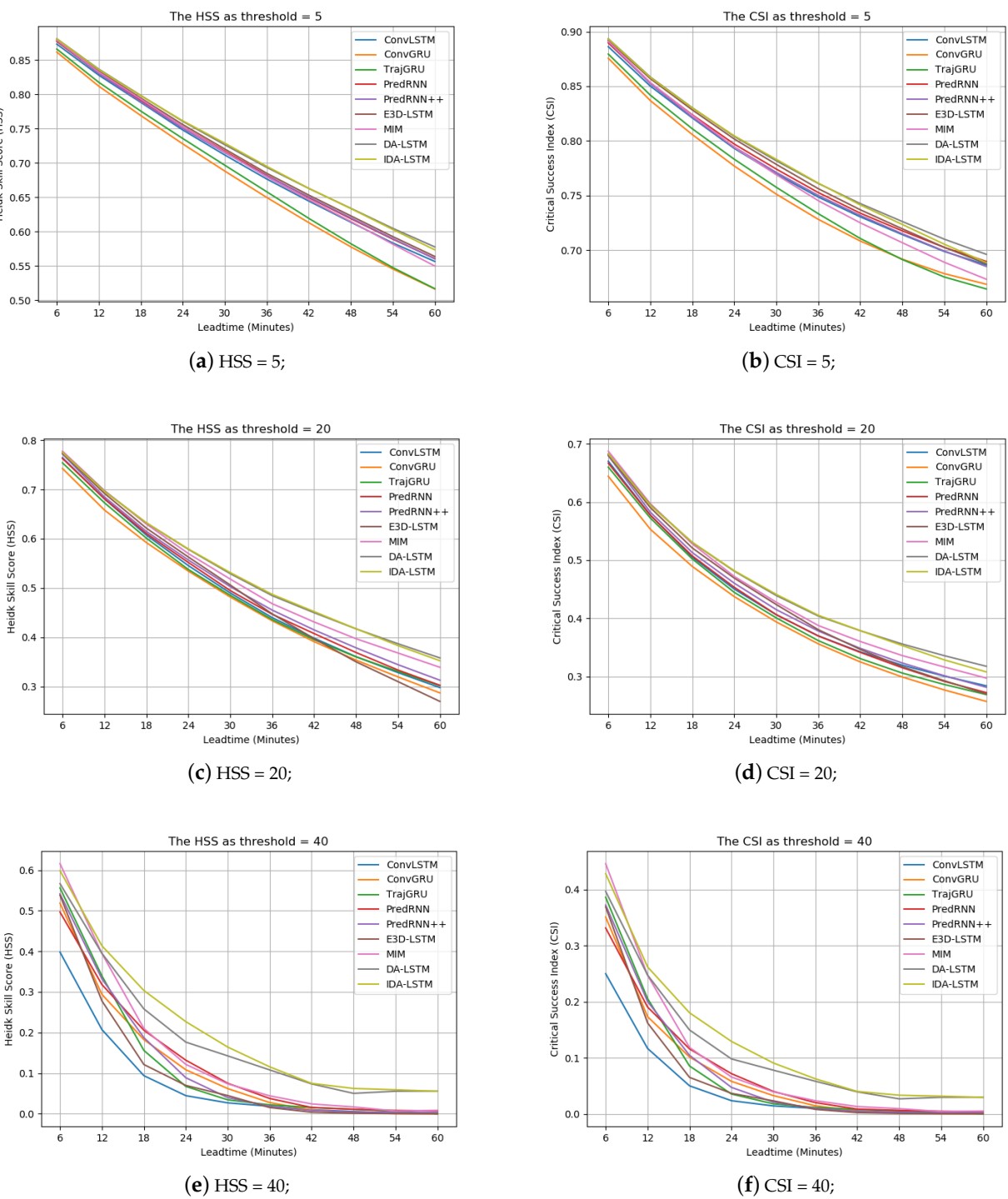

**Figure 10.** The performance changes against different nowcasting lead time in terms of HSS and CSI scores. (Best view in color).

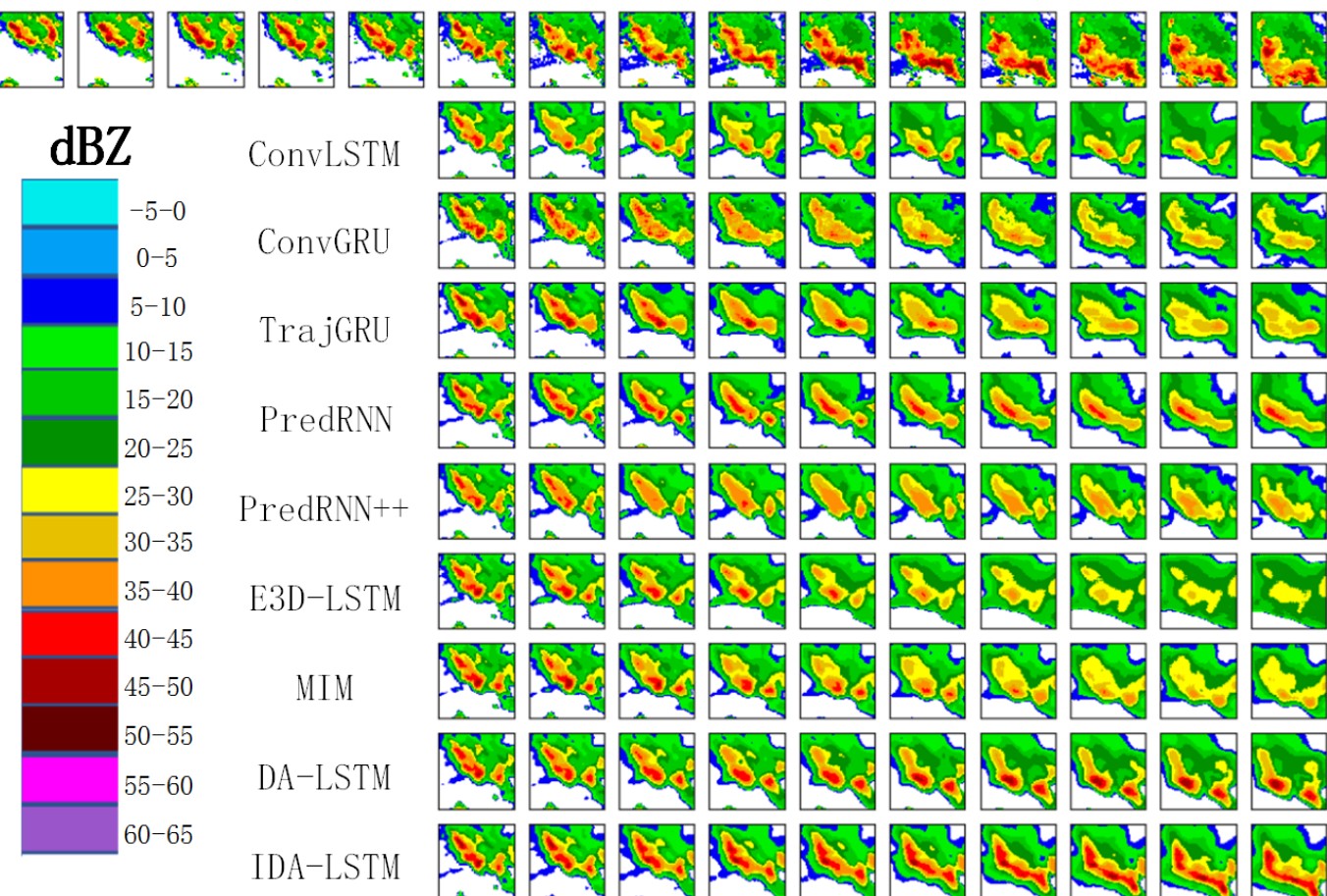

**Figure 11.** The prediction results of all methods on an example from the CIKM AnalytiCup 2017 competition. The first five images in the first row are the input, and the remainders denote the ground-truth output. Other rows are the prediction of various models.

### 3.3. Ablation Study

To further investigate the effectiveness of the two proposed mechanisms, we conduct ablation study in this subsection.

**Context Interaction:** To validate the effectiveness of the proposed interaction framework, we embed it into the ConvLSTM, PredRNN, PredRNN++ and DA-LSTM, respectively and test their performances. Table 3 shows the results of these models without the interaction and with different number of interactions. The methods with the prefix "i" denote the ones with interactions and the superscript number denotes interaction times. We can see that when equipped with interactions, the performances of these models all improve. The improvement is especially obvious at 40 dBZ threshold. The fact demonstrates the effectiveness of the proposed interaction framework.

Similarly, we also depict the HSS and CSI curves w.r.t. different nowcasting lead times at 40 dBZ threshold in Figures 12 and 13, respectively. The methods with the prefix "i" denote the ones with the interactions. We observe that the performance with the interactions is better at different nowcasting lead times. The fact further validates the effectiveness of the developed interaction framework.

**Table 3.** Comparison results with the different number of interactions on the CIKM AnalytiCup 2017 competition dataset in terms of HSS and CSI. Bold denotes the best evaluate index among all models.

| dBZ Threshold | HSS ↑ | | | | CSI ↑ | | | | MAE ↓ | SSIM ↑ |
|---|---|---|---|---|---|---|---|---|---|---|
| | 5 | 20 | 40 | avg | 5 | 20 | 40 | avg | | |
| ConvLSTM | 0.7035 | 0.4819 | 0.1081 | 0.4312 | 0.7656 | 0.4034 | 0.0578 | 0.4089 | 15.06 | 0.2229 |
| IConvLSTM [1] | **0.7149** | 0.4889 | 0.1236 | **0.4424** | 0.7769 | 0.4119 | 0.0667 | 0.4184 | 14.62 | **0.3390** |
| IConvLSTM [2] | 0.7055 | **0.5001** | 0.1215 | 0.4424 | 0.7668 | **0.4120** | 0.0652 | 0.4146 | **14.42** | 0.3365 |
| IConvLSTM [3] | 0.7092 | 0.4740 | **0.1247** | 0.4360 | **0.7784** | 0.4118 | **0.0671** | **0.4191** | 15.11 | 0.3372 |
| IConvLSTM [4] | 0.5645 | 0.4044 | 0.0830 | 0.3503 | 0.6305 | 0.3362 | 0.0453 | 0.3373 | 20.65 | 0.3111 |
| IPredRNN | 0.7081 | 0.4911 | 0.1558 | 0.4516 | 0.7691 | 0.4048 | 0.0854 | 0.4198 | 14.54 | 0.3341 |
| IPredRNN [1] | **0.7133** | 0.5108 | 0.2047 | **0.4762** | 0.7685 | 0.4188 | 0.1151 | 0.4341 | **14.03** | **0.3488** |
| IPredRNN [2] | 0.7081 | 0.5039 | 0.1531 | 0.4550 | 0.7710 | 0.4154 | 0.0836 | 0.4233 | 14.40 | 0.3312 |
| IPredRNN [3] | 0.7001 | **0.5179** | 0.1951 | 0.4710 | 0.7710 | **0.4289** | 0.1089 | **0.4359** | 14.52 | 0.3281 |
| IPredRNN [4] | 0.7111 | 0.5019 | **0.2155** | 0.4762 | **0.7726** | 0.4101 | **0.1218** | 0.4348 | 14.20 | 0.3327 |
| IPredRNN++ | 0.7075 | 0.4993 | 0.1575 | 0.4548 | 0.7670 | 0.4137 | 0.0862 | 0.4223 | 14.51 | 0.3357 |
| IPredRNN++ [1] | **0.7188** | **0.5100** | 0.2004 | 0.4764 | 0.7759 | **0.4251** | 0.1124 | 0.4378 | **14.13** | **0.3513** |
| IPredRNN++ [2] | 0.7119 | 0.5037 | 0.2098 | 0.4751 | 0.7715 | 0.4204 | 0.1181 | 0.4367 | 14.33 | 0.3423 |
| IPredRNN++ [3] | 0.7023 | 0.4995 | 0.1610 | 0.4543 | 0.7665 | 0.4110 | 0.0882 | 0.4219 | 14.59 | 0.3255 |
| IPredRNN++ [4] | 0.7153 | 0.4968 | **0.2172** | **0.4764** | **0.7774** | 0.4239 | **0.1234** | **0.4416** | 14.62 | 0.3487 |
| DA-LSTM | **0.7185** | 0.5251 | 0.2127 | 0.4854 | 0.7765 | **0.4376** | 0.1202 | 0.4448 | 14.10 | 0.3479 |
| IDA-LSTM [1] | 0.7093 | 0.5065 | 0.1606 | 0.4588 | 0.7683 | 0.4218 | 0.0881 | 0.4261 | 14.38 | 0.3345 |
| IDA-LSTM [2] | 0.7179 | **0.5264** | **0.2262** | **0.4902** | 0.7752 | 0.4372 | **0.1287** | **0.4470** | **14.09** | **0.3506** |
| IDA-LSTM [3] | 0.7179 | 0.5156 | 0.1879 | 0.4738 | **0.7798** | 0.4342 | 0.1044 | 0.4395 | 14.18 | 0.3436 |
| IDA-LSTM [4] | 0.7068 | 0.5085 | 0.1930 | 0.4694 | 0.7631 | 0.4125 | 0.1081 | 0.4279 | 14.21 | 0.3461 |

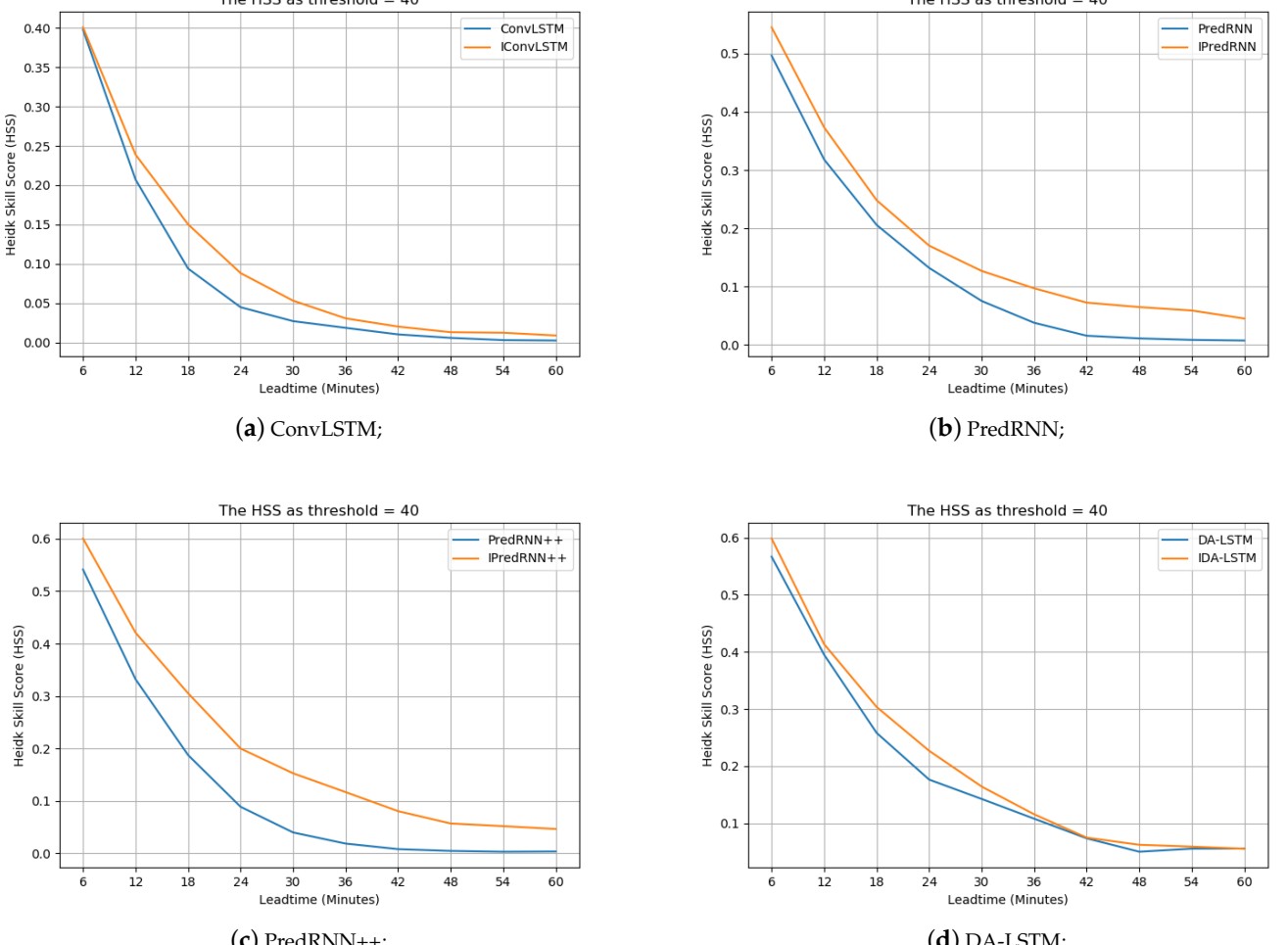

(**a**) ConvLSTM;

(**b**) PredRNN;

(**c**) PredRNN++;

(**d**) DA-LSTM;

**Figure 12.** The performance changes against different nowcast lead times in interaction ablation study in terms of HSS as the threshold is 40 dBZ. (Best view in color).

**(a)** ConvLSTM;

**(b)** PredRNN;

**(c)** PredRNN++;

**(d)** DA-LSTM;

**Figure 13.** The performance changes against different nowcast lead times in interaction ablation study in terms of CSI as the threshold is 40 dBZ. (Best view in color).

To visually compare the results with/without the interactions of the four methods ConvLSTM, PredRNN, PredRNN++, IDA-LSTM, we depict in Figure 14 four groups of prediction results. We can see that the models with the interactions predict the high echo value regions (red parts) better than their counterparts without the interactions. The results further demonstrate the superiority of the interaction framework.

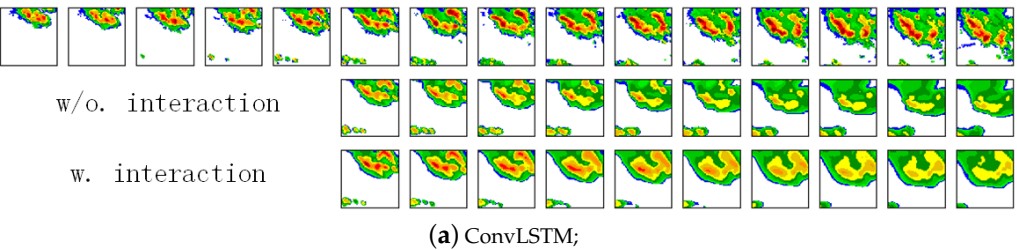

**(a)** ConvLSTM;

**Figure 14.** *Cont.*

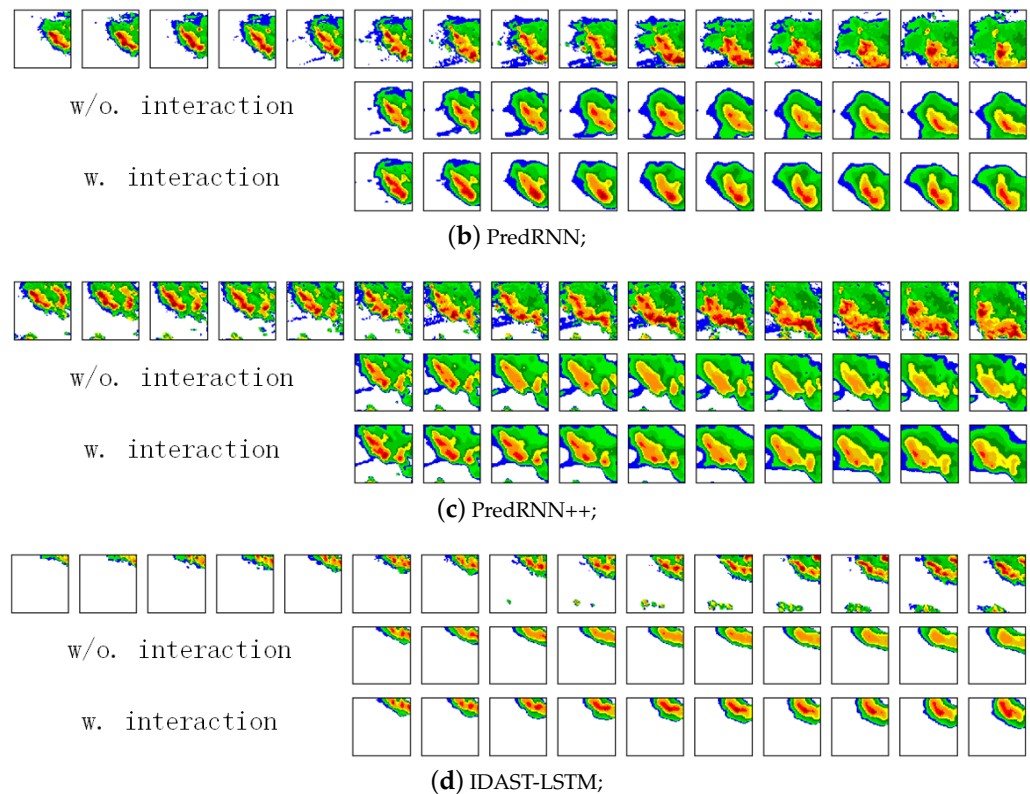

**Figure 14.** The four group interaction ablation examples from the CIKM AnalytiCup 2017 competition. These groups are ConvLSTM, PredRNN, PredRNN++, and DAST-LSTM respectively from top to bottom. The first five images in the first row are the input, and the remainders denote the ground-truth output. Other rows are the prediction of various models. (Best view in color).

**Dual Attention Module:** The purpose of the dual attention module is to exploit more spatiotemporal representation from huge history temporal memories and to preserve adequate information involving high echo value regions. To validate its effectiveness, we embed different types of attention schemes into the PredRNN model. The attention schemes include "without attention", "with spatial attention", "with channel attention" and "with the dual attention". Table 4 shows the results of the four schemes. Here, CA-LSTM and SA-LSTM denote the PredRNN model with channel attention and spatial attention respectively. The DA-LSTM represents the PredRNN model with our dual attention. We observe from the table that the DA-LSTM delivers the best results, followed by CA-LSTM and SA-LSTM. The PredRNN method without any attention performs the worst. The observations validate the effectiveness of the attention schemes and the superiority of our dual attention mechanism.

**Table 4.** Comparison results of attention mechanism ablation study on the CIKM AnalytiCup 2017 competition dataset in terms of HSS and CSI. Bold denotes the best evaluate index among all models.

| Model | HSS ↑ | | | | CSI ↑ | | | | MAE ↓ | SSIM ↑ |
|---|---|---|---|---|---|---|---|---|---|---|
| | 5 | 20 | 40 | avg | 5 | 20 | 40 | avg | | |
| PredRNN | 0.7081 | 0.4911 | 0.1558 | 0.4516 | 0.7691 | 0.4048 | 0.0854 | 0.4198 | 14.54 | 0.3341 |
| SA-LSTM | 0.7042 | 0.4982 | 0.1481 | 0.4502 | 0.7689 | 0.4143 | 0.0808 | 0.4213 | 14.68 | 0.3241 |
| CA-LSTM | 0.7115 | 0.5066 | 0.1575 | 0.4585 | 0.7733 | 0.4172 | 0.0861 | 0.4255 | 14.23 | 0.3296 |
| DA-LSTM | **0.7185** | **0.5251** | **0.2127** | **0.4854** | **0.7765** | **0.4376** | **0.1202** | **0.4448** | **14.10** | **0.3479** |

Again, we show the HSS and CSI curves of the four methods against different now-casting lead times in Figure 15. It can be seen that the DA-LSTM equipped with our dual

attention mechanism consistently performs the best. Moreover, as the threshold increases from 5 dBZ to 40 dBZ, the improvement of our method becomes more and more obviously. The observation demonstrates that the developed dual attention mechanism is especially helpful for the prediction of high echo value parts.

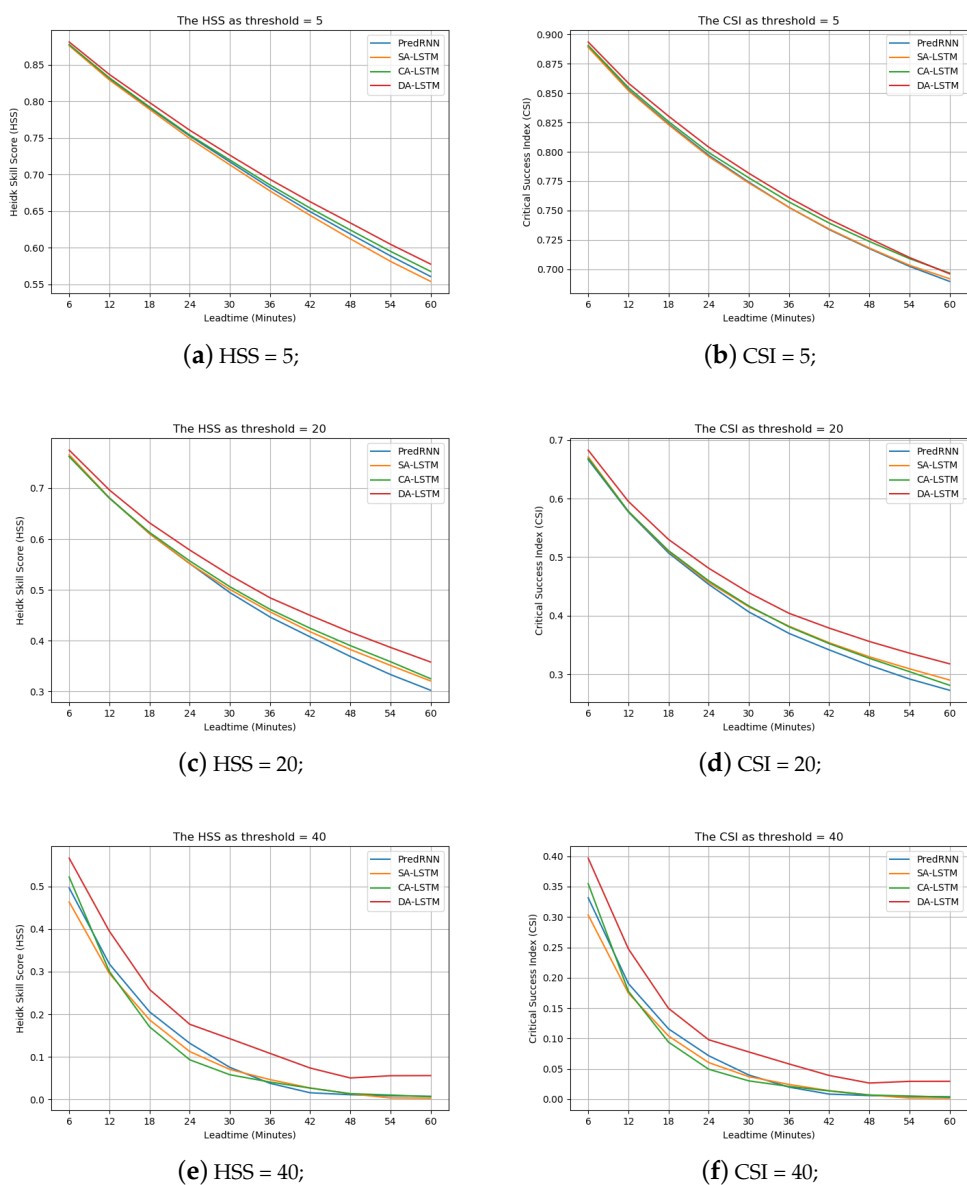

**Figure 15.** The performance changes against different nowcast lead times in attention mechanisms ablation study in terms of HSS and CSI scores. (Best view in color).

Figure 16 shows the one-hour prediction echo maps of the four schemes on an example. We can see that the method with our dual attention mechanism can better preserve the high echo value parts (red parts) than the one with single attention or without attention. The result further validates the effectiveness of the developed dual attention mechanism.

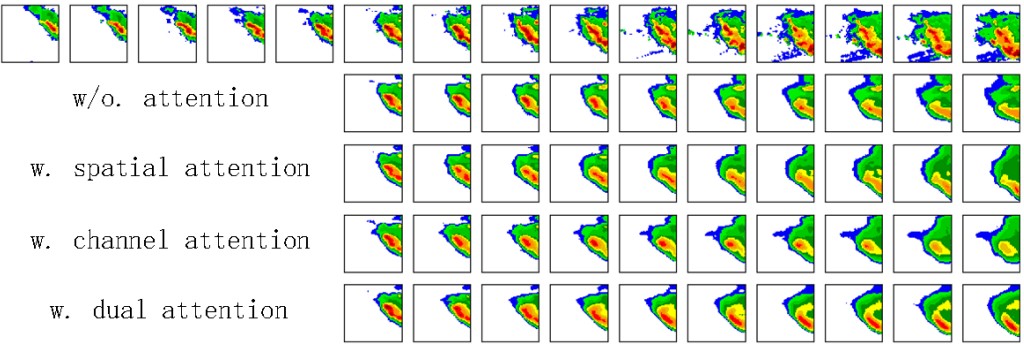

**Figure 16.** The attention mechanism ablation example from the CIKM AnalytiCup 2017 competition. The first five images in the first row are the input, and the remainders denote the ground-truth output. Other rows are the prediction of various models. (Best view in color).

## 4. Conclusions

In this paper, we propose a novel radar echo map extrapolation method, namely, IDA-LSTM. In the method, an interaction framework is developed for the ConvRNN unit to fully exploit the short-term context information, which can be applied in any ConvRNN based model. The ablation study shows that it can improve the prediction almost in all ConvRNN models after several interactions. Moreover, a dual attention mechanism is developed to combine the channel attention and spatial attention, which can recall forgotten information in ConvRNN to model long-term dependency. The experiment shows that embedding dual attention into the PredRNN achieve better performance. By combining the two improvements, we proposed IDA-LSTM which overcomes the underestimation issue of high echo value parts that existing deep learning extrapolation methods suffer from. By comparing other existing algorithms, the superiority and performance of it have been fully demonstrated. In the future, we will explore the prediction of radar echo maps at multiple heights to improve the nowcasting in some extreme weather events such as convection.

**Author Contributions:** Conceptualization, X.L. and C.L.; methodology, C.L.; software, C.L. and Y.W.; validation, C.L. and X.Z.; formal analysis, C.L.; investigation, Y.Y.; resources, Y.Y.; data curation, Y.W.; writing—original draft preparation, C.L.; writing—review and editing, Y.Y., X.L. and X.Z.; visualization, Y.W.; supervision, Y.Y., X.L.; project administration, X.L.; funding acquisition, Y.Y. All authors have read and agreed to the published version of the manuscript.

**Funding:** This work was supported in part by the Shenzhen Science and Technology Program under Grant JCYJ20180507183823045 and JCYJ20200109113014456.

**Institutional Review Board Statement:** The study did not involve humans or animals.

**Informed Consent Statement:** The study did not involve humans.

**Data Availability Statement:** The CIKM AnalytiCup 2017 Dataset can be download according to this address: https://tianchi.aliyun.com/dataset/dataDetail?dataId=1085.

**Conflicts of Interest:** The funders had no role in the design of the study; in the collection, analyses, or interpretation of data; in the writing of the manuscript, or in the decision to publish the results.

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
