# Peer review of "A Novel LSTM Model with Interaction Dual Attention for Radar Echo Extrapolation"

_remotesensing, doi:10.3390/rs13020164_

Round 1

Reviewer 1 Report

The present paper is concerned with the improvement of deep-learning based radar nowcasting. The paper is well-written and interesting to read, and I have only a few minor comments that the authors may wish to take into consideration before submitting a final version of their paper:

1) How are the scores defined that the authors use for verification? While the scores presented are all scalars, they could be defined pointwise for all pixels of the domain, i.e. they could also be spatially dependent. Are the authors comparing their results against the ground truth and then averaging the pointwise spatial results to provide aggregate results?

2) While the short-term predictions are quite impressive, I am not sure that for the longer-term predictions (say, 30 minutes and longer), a deep-learning based strategy on two-dimensional radar data alone will be the most promising. Fundamentally, the two-dimensional radar data do not capture the full three-dimensional evolution of extreme weather events such as convection, which would require the vertical channel as well. As such, it would be interesting to see the methodology of the authors applied to the fully three-dimensional radar data to see if that would improve the longer-term predictions even further. I understand that this would require substantial additional work, so I propose this be done in a future, separate work. The authors may still want to comment on this issue here.

Reviewer 2 Report

An extensive study of the training part is totally missing. Especially when dealing with deep learning, this must be done for reproducibility. Moreover, the experiment section should study others evaluations metrics to show the improvement of the method of the authors among others.  

Reviewer 3 Report

This paper introduces an interesting approach to improve Radar Echo Extrapolation using interaction Interaction Dual Attention LTSTM.

The paper is well written and modify a recent method (Dual Attention LSTM).

Some drawbacks in this paper:

  1. The detailed information for The interaction block is nowhere to be found (Number of filters, filter size, stride, etc).
  2. On page 15, spatial attention module, the writers reshape the feature map from R^N×C×H×W to R^N×(H∗W)×C. By vectorizing the Height and Width the writers are losing spatial information. Please clarify this step.
  3. Please provide a subchapter that explains the dataset in detail, then relate it with the method chosen. 
  4. The conclusion should be improved because of many unconcluded experiments.
  5. Based on the experiment result in table 3, there is not too much improvement between the older methods with Interaction vs the newly proposed method. Please reevaluate the abstract that say IDA-LSTM is superior to other methods. 

Reviewer 4 Report

In this work a new neural network architecture IDA-LSTM is proposed and tested for processing weather radar data. The idea of the architecture looks promising and arouses some interest. However, the research method leaves some questions open.

  1. Why didn't the authors use a more understandable criterion of the average discrepancy between the actual and predicted images to compare the performance of different neural networks?
  2. As far as can be understood the use of thresholding the initial images was to demonstrate the ability of IDA-LSTM architecture to best reproduce the "extreme" features of the images. However, the question arises: is it correct to directly compare this architecture with other implementations that did not explicitly pose the same problem? Wouldn't it be more correct then to train all the neural networks on images obtained from initial ones using thresholds at given levels, and only then compare which of the neural networks best predicts the evolution of regions with extreme values?
  3. How can the authors comment on the fact (see Table 3) that the IPredRNN++1 architecture with only one interaction demonstrated the absolutely best performance according to the HSS criterion, and among the DA-LSTM architectures the best result, inferior to IPredRNN++1, was shown by the scheme without interactions.
  4. Why are there "four" examples mentioned in the captions to Figures 3 and 6, while the figures themselves have 8 columns each? Does a column refer to an individual example, or are they paired in some way?

I believe that in order to improve clarity the authors need to provide clear answers to these questions. In this case the paper can be recommended for publication.

Round 2

Reviewer 2 Report

The authors have greatly improved the paper. So I recommand acceptation for this paper.